# Cold Atmospheric Pressure Plasma-Activated Medium Modulates Cellular Functions of Human Mesenchymal Stem/Stromal Cells In Vitro

**DOI:** 10.3390/ijms25094944

**Published:** 2024-05-01

**Authors:** Olga Hahn, Tawakalitu Okikiola Waheed, Kaarthik Sridharan, Thomas Huemerlehner, Susanne Staehlke, Mario Thürling, Lars Boeckmann, Mareike Meister, Kai Masur, Kirsten Peters

**Affiliations:** 1Institute of Cell Biology, Rostock University Medical Center, 18057 Rostock, Germany; olga.hahn@med.uni-rostock.de (O.H.); okikiolawaheed@gmail.com (T.O.W.); kaarthikgauthum@gmail.com (K.S.); thomas.huemerlehner@gmail.com (T.H.); susanne.staehlke@med.uni-rostock.de (S.S.); 2Microfluidics, Faculty of Mechanical Engineering and Marine Technology, University of Rostock, 18059 Rostock, Germany; mario.thuerling@uni-rostock.de; 3Clinic and Polyclinic for Dermatology and Venerology Rostock, Rostock University Medical Center, 18057 Rostock, Germany; lars.boeckmann@med.uni-rostock.de; 4Leibniz-Institute for Plasma Science and Technology e.V., 17489 Greifswald, Germany; mareike.meister@inp-greifswald.de (M.M.); kai.masur@inp-greifswald.de (K.M.)

**Keywords:** cold atmospheric pressure plasma, plasma-activated medium, mesenchymal stem/stromal cells, oxidative stress, metabolic activity, cell migration

## Abstract

Cold atmospheric pressure plasma (CAP) offers a variety of therapeutic possibilities and induces the formation of reactive chemical species associated with oxidative stress. Mesenchymal stem/stromal cells (MSCs) play a central role in tissue regeneration, partly because of their antioxidant properties and ability to migrate into regenerating areas. During the therapeutic application, MSCs are directly exposed to the reactive species of CAP. Therefore, the investigation of CAP-induced effects on MSCs is essential. In this study, we quantified the amount of ROS due to the CAP activation of the culture medium. In addition, cell number, metabolic activity, stress signals, and migration were analyzed after the treatment of MSCs with a CAP-activated medium. CAP-activated media induced a significant increase in ROS but did not cause cytotoxic effects on MSCs when the treatment was singular and short-term (one day). This single treatment led to increased cell migration, an essential process in wound healing. In parallel, there was an increase in various cell stress proteins, indicating an adaptation to oxidative stress. Repeated treatments with the CAP-activated medium impaired the viability of the MSCs. The results shown here provide information on the influence of treatment frequency and intensity, which could be necessary for the therapeutic application of CAP.

## 1. Introduction

Cold atmospheric pressure plasma (CAP) has attracted increasing interest in biomedical research in recent years since it offers a wide range of therapeutic options [1]. CAP treatment induces a complex mixture of compounds such as different reactive oxygen species (ROS) and nitrogen species (RNS), charged atoms and molecules, electrons, ultraviolet radiation, and electromagnetic fields, which may exert synergistic effects in a biological environment [2]. Compounds of the ROS class, such as superoxide radicals (O_2_^•−^), hydrogen peroxide (H_2_O_2_), and hydroxyl radicals (-OH), are highly reactive. Physiologically, ROS are formed as by-products of cell metabolism, particularly in the mitochondria [3]. If the ratio between the production of ROS and the cellular defense mechanisms is out of balance, the resulting situation is referred to as oxidative stress. It is known that various ROS can be produced by CAP, some of which cause oxidative stress in cells [4,5]. The cellular antioxidant defense mechanisms (e.g., proteins such as superoxide dismutase/SOD or catalase) serve to neutralize ROS and, thus, protect the cells from oxidative stress. Cell components such as lipids, proteins, and DNA can be damaged via oxidative stress. This oxidative stress, in turn, can trigger inflammatory processes and, thus, for example, neurodegenerative and cardiovascular diseases [6]. The range of therapeutic applications for CAP extends from decontamination procedures and the surface treatment of implants to its use in dentistry, plastic surgery, and skin disease treatments [7,8,9,10,11,12]. CAP is seen as a potential therapeutic trigger, particularly in regenerative medicine and wound healing, as CAP can be used to modulate specific cellular processes [13]. Consequently, any signaling pathway regulated by or associated with ROS can be influenced directly or indirectly by CAP treatment [5].

Mesenchymal stem/stromal cells (MSCs) are adult stem cells that play a central role in tissue homeostasis and regeneration due to their versatile capabilities. Their most distinctive property is their ability to differentiate into various, mainly mesenchymal, cell types, including osteoblasts, chondrocytes, and adipocytes. This differentiation capacity allows MSCs to regenerate and maintain various tissues and organs [14,15]. MSCs further make a significant contribution to tissue regeneration through the production of various growth factors. These factors influence the proliferation and differentiation of neighboring cells, thereby promoting the healing of injured tissue [16,17]. MSCs can also modulate the extracellular matrix (ECM), crucial in maintaining tissue integrity and function [18,19]. The migratory and homing activities of MSCs are also of particular interest in wound healing.

MSCs can specifically migrate to sites of injury, emphasizing their involvement in the initial stages of wound healing [20,21,22]. Furthermore, they also have a particularly strong antioxidant activity that helps to protect cells from oxidative stress, which could be a crucial aspect in their cell therapeutic application [23,24]. In addition, the MSCs may have immunoregulatory and pro-angiogenic properties [25,26]. In summary, MSCs, through their versatility and functions, are pivotal players in tissue regeneration, immunomodulation, and wound healing. In particular, it must be emphasized that during the treatment of open wounds with CAP, the MSCs of the adipose tissue (adipose tissue-derived MSCs/adMSCs) are directly exposed to the reactive components of the CAP. In this context, investigating adMSC response to CAP treatment is of great interest. In our study, we aimed to elucidate different aspects of the responses of human adMSCs to indirect CAP treatment using a CAP-activated cell culture medium (CAP-actM). For this purpose, a plasma source was selected, which was characterized concerning the potential for forming ROS and RNS, among other things [27]. In particular, we focused on the viability, energy metabolism, cellular stress responses, and migration activity of human adMSCs induced using a CAP-activated cell culture medium. A deeper understanding of these aspects could lead to fundamental insights into the cellular mechanisms and contribute to possible application strategies in wound healing therapy via CAP.

## 2. Results

### 2.1. Quantification of ROS in CAP-Activated Medium and Cultured adMSCs

To determine the extent of the resulting oxidative stress, the amount of H_2_O_2_ in the medium without and with CAP activation was quantified immediately after CAP exposure. The cell culture medium was CAP-activated by exposing the medium to the plasma source for 1 min (CAP-actM^1min^), 3 min (CAP-actM^3min^), and 5 min (CAP-actM^5min^) (i.e., CAP exposure time of the medium). This CAP medium exposure led to a significant increase in the H_2_O_2_ concentration in the medium, depending on the exposure time (Figure 1A). In principle, the H_2_O_2_ concentration was higher in the absence of pyruvate (i.e., DMEM^w/o pyr^). For example, after 1 min of CAP exposure, the initial H_2_O_2_ concentration was a median of 96 µM of H_2_O_2_. A 5 min exposure resulted in a 4.5-fold increase with a concentration of 435 µM of H_2_O_2_. Twenty-four hours after CAP exposure, the H_2_O_2_ concentration in the medium was reduced to 57% of this initial value (for 1 min with 56 µM of H_2_O_2_ and 5 min with 248 µM of H_2_O_2_; Figure 1A).

The intracellular level of ROS was then determined by treating adMSCs with a CAP-activated cell culture medium. In the presence of pyruvate, a significant increase in ROS level was only observed when cells were treated with the 5 min CAP-activated medium. Interestingly, the treatment of the adMSCs with the 1 and 3 min CAP-activated media resulted in a slight but significant decrease in the level of ROS (Figure 1B). This reduction in ROS amount already indicates that the cells can decompose and metabolize the H_2_O_2_ present with the help of intracellular antioxidant components. In the absence of the ROS scavenger sodium pyruvate, there was a CAP treatment time-dependent significant increase in the relative intracellular level of ROS in the adMSCs (Figure 1B). 

### 2.2. Analysis of Cell Viability, Energy Metabolism, and Intracellular Calcium Levels in adMSCs upon Treatment with CAP-Activated Medium

To determine the potential effect of CAP-activated medium on adMSCs in vitro in terms of cell number and metabolic activity, adMSCs were cultivated in the presence of differently prepared media for up to 7 days. In this procedure, the medium was changed every second day so that the cell cultures were treated with the CAP-activated medium either once (for 1 day), twice (for 3 days), or thrice (for 7 days). There were no effects on cell number and metabolic activity after 1 day with a single treatment (Figure 2A,B). In contrast, after 3 days of cultivation in the presence of CAP-activated medium (double treatment), a significantly lower number of adMSCs was observed with 3 and 5 min CAP-activated media (reduction of approx. 7 and 10%, respectively; Figure 2C). After 7 days of cultivation in the presence of CAP-activated medium with triple treatment, the number of adMSCs further decreased by 16% and 21% with the 3 and 5 min activated CAP media (Figure 2E). Despite the decrease in cell number after 3 days, adMSCs showed no differences in metabolic activity as determined with the MTS conversion assay (Figure 2D). After 7 days of cultivation, the metabolic activity correlated with the decreasing cell number (Figure 2F), indicating an impairment of cell viability. In the presence of the ROS scavenger sodium pyruvate, no change in cell number and metabolic activity was observed.

Based on these data, the analyses of mitochondrial respiration were then carried out. In contrast to the preceding MTS test, 1 day of cultivation in a CAP-activated medium already led to a significant decrease in maximum respiration (Figure 3A). However, this decrease was no longer detectable after 3 days with a double medium change and, thus, double treatment with CAP-activated medium (Figure 3C), indicating a possible adaption of the cells. The basal respiration, ATP-coupled respiration, and non-mitochondrial respiration were not affected by cultivation with a CAP-activated medium for up to 3 days. Furthermore, no shift in glycolytic metabolism was induced via the treatment of adMSCs with a CAP-activated medium.

In addition, the calcium signaling was analyzed by detecting the basal intracellular calcium (Ca^2+^) level. A significant increase in intracellular Ca^2+^ was observed here, induced with a CAP-activated medium after 1 and 3 days of cultivation (Figure 3B,D). This effect could also be demonstrated in the presence of sodium pyruvate, albeit to a much lesser extent.

### 2.3. Cell Stress-Associated Protein Expression upon Treatment with CAP-Activated Medium

Further analyses included the quantification of cell stress-associated proteins. The treatment of adMSCs to CAP-activated medium for 1 day resulted in a cell stress-associated response in 24 proteins (out of 26 total proteins tested; Table A1 and Figure A2). In particular, treatment with the short-term CAP-activated medium (CAP-actM^1min^) led to a significant increase in 11 proteins (Figure 4). These proteins were associated with oxidative stress, such as carbonic anhydrase, paraoxonase 2 (PON2), superoxide dismutase 2 (SOD2), and thioredoxin-1 in the adMSCs. In addition, transcription factors involved in cytoprotection, such as cited-2, heat shock protein 70 (HSP70), and sirtuin 2, were also increased in response to treatment with short-term CAP-activated medium (CAP-actM^1min^).

Thus, while treatment with the short-term CAP-activated medium with relatively low levels of H_2_O_2_ and ROS (i.e., CAP-actM^1min^) had this enhancing effect on cell stress-associated proteins, treatment with the longer-term CAP-activated media (i.e., CAP-actM^5min^), with relatively higher levels of H_2_O_2_ and ROS, resulted in a significant reduction in these cell stress-associated proteins (Figure 4). This reduction indicates a possible depletion of the antioxidant capacity following treatment to this increased oxidative stress.

### 2.4. Determination of the Migratory Activity of adMSCs after Treatment with CAP-Activated Medium

MSC migration is a relevant feature during tissue regeneration. Therefore, we analyzed to what extent the migration of adMSCs was influenced by treatment with CAP-activated medium using the so-called scratch assay (also referred to as wound closure assay). For this assay, a scratch is made in a monolayer of adherent-growing cells, and the closure of the wound area over time is recorded as a measure of cell migration. Therefore, microscopic images captured after fluorescence staining were subjected to software-assisted image analysis. The treatment of adMSCs with a CAP-activated medium resulted in increased cell coverage in the wound area compared to untreated cell cultures (Figure 5A). This increase was dependent on the duration of the CAP treatment (the longer and, thus, the more intensively, the medium was CAP activated, the more overgrown the wound area became). Software-assisted quantification revealed a 14% increase in adMSC migration after a single CAP treatment with a 3 min and a significant increase of 36% after a 5 min treatment with CAP-activated media after one day (Figure 5B). This increased migration rate was also observed after 3 days with a double treatment with CAP-activated medium, albeit to a lesser extent: the 5 min CAP treatment led to a significant increase of about 25% (Figure A1). The statistical analyses showed that the intensity of the CAP activation of the medium was the decisive parameter compared to the frequency of the treatment of the cells with CAP-activated medium (*p* < 0.0001 vs. *p* = 0.6301). The effects were significantly lower in the presence of the ROS scavenger sodium pyruvate (Figure A1).

## 3. Discussion

CAP treatment can generate reactive species such as ROS and RNS [13], influencing specific biological reactions [4]. Although the effects of plasma-induced oxidative stress have been extensively studied [5], many questions still need to be answered due to the complex interactions between numerous influencing factors and the heterogeneity of the reported results. Therefore, studies to understand the complex interactions of plasma-induced oxidative stress and the effects of plasma on the regenerative capacity and survival of MSCs might be essential for therapeutic application.

### 3.1. Oxidative Stress Induced via CAP-Activated Medium

Oxidative stress occurs when the amount of ROS exceeds the cell’s antioxidant defense mechanisms [28]. Oxidative stress is caused either by the ROS concentration exceeding a certain level or by the cell’s antioxidant defense mechanisms being too low. Both parameters are influenced by ambient conditions, such as the redox status of the cells or tissue type affected. Oxidative stress can cause cellular dysfunction and cell death [29].

The cytotoxicity of the direct plasma treatment of cells is well known. Still, the indirect plasma contact of cells via the cell culture medium also induces structural changes and cytotoxicity due to the formation of ROS and other reactive molecules, such as the biologically active peroxynitrous acid [30,31,32,33,34]. Our experiments revealed an increased H_2_O_2_ level in the cell culture medium after CAP activation, which decreased significantly after 24 h. This significant decrease in the H_2_O_2_ concentration in the medium could be due to various aspects, such as the pH value of the medium, the ambient temperature, or the presence of catalytically active compounds. For example, a pH value above pH 5 leads to a significant decomposition of H_2_O_2_. Since the pH of the buffered cell culture medium is between 7.0 and 7.6, it can be concluded that the pH of the (CAP-activated) medium contributed to H_2_O_2_ degradation. Although Mann et al. have demonstrated a reduction in pH from pH 7 to pH 6 in a non-buffered saline solution [27], it can be assumed that the pH of the buffered cell culture medium did not change significantly in our experiments, as the color of the medium did not change after the CAP treatments. In addition, the degradation of H_2_O_2_ can be triggered by the presence of, e.g., metal ions such as iron, copper, manganese, nickel, and chromium, as well as by specific enzymes (e.g., catalase) in the medium and serum [35].

The increased H_2_O_2_ content in the CAP-activated cell culture medium was accompanied by increased intracellular ROS in adMSCs (especially without the ROS scavenger pyruvate). However, this increase did not lead to cytotoxic effects after 1 day with a single treatment. This result suggests that the adMSCs can efficiently degrade and metabolize the externally supplied ROS (H_2_O_2_ among them) with the help of intracellular antioxidant components such as glutathione or enzymes such as SOD (as already shown for MSCs from adipose tissue and bone marrow) when the treatment is carried out once and for a short time. In contrast, longer-term treatment with a CAP-activated medium (up to 7 days) and repeated medium changes impaired the viability of the adMSCs. This impairment suggests that the 7-day treatment in combination with the repeated application of CAP-activated media may have exceeded the cells’ ability to compensate for oxidative stress. Due to the relatively low H_2_O_2_ concentrations in the medium after the 1 and 3 min CAP activations, we refer to the resulting CAP media as mildly activated in the following section. In contrast, we refer to the 5 min CAP treatment with relatively high H_2_O_2_ concentrations as strongly activated.

Experimental evidence suggests that the critical, cytotoxic compound H_2_O_2_ is the main reactive species produced via CAP treatment, resulting in apoptosis and cell death by generating OH radicals within the cells via the Haber–Weiss reaction [36]. Mann et al., who also used the specific plasma source from our study, also showed that the H_2_O_2_ generated via the plasma treatment accounted for most of the reactive components. In contrast, the reactive nitrate (NO_3_^−^) could be detected in the aqueous solution regardless of the plasma treatment and the concentration of reactive nitrite (NO_2_^−^) was about ten times lower than that of H_2_O_2_ [27]. The constant, unaltered cellular metabolic activity after a single CAP treatment in our experiments (with approx. 500 μM H_2_O_2_) is, therefore, due to the high capacity of the MSCs to rapidly metabolize the H_2_O_2_ through their robust antioxidant defense mechanisms [23]. Zenin et al. [37] hypothesize that under oxidative stress conditions, it is not the H_2_O_2_ concentration per se that is the damaging factor but the total dose of H_2_O_2_ to which the cell is exposed and which it can decompose.

The cytotoxic effect of H_2_O_2_ has been demonstrated in several cell lines through DNA strand breaks and damage to other cellular components [30,38,39]. H_2_O_2_ concentrations above 100 μM were severely cytotoxic to the mammalian cell line CHO-K1 [40]. Shojaei et al. showed the induction of necrosis and apoptosis as well as changes in cell cycle regulation after 48 h when adMSCs were treated with a strongly activated CAP medium [41]. Despite detailed investigations, no increased adMSC proliferation could be detected in our experimental setup. This finding contrasts the results of previous studies, which showed an increase in proliferation, specifically in adMSCs [41], while other cells analyzed, e.g., HeLa cells, underwent apoptosis [42]. In addition, previous studies have shown that the cell culture medium composition significantly influences the results obtained through CAP activation [43]. For example, a strong protective effect of fetal bovine serum was demonstrated in the direct CAP treatment of mammalian cells, presumably through scavenging reactive species such as H_2_O_2_ [43]. Therefore, Boehm et al. analyzed CAP-treated fetal bovine serum samples for persistent cytotoxic activity [40]: the authors hypothesize that more stable compounds, such as lipid peroxides, are formed in CAP-treated sera, which may contribute to cell damage. Furthermore, there is evidence that oxidative stress accumulates misfolded proteins and triggers the unfolded protein response, a cellular mechanism to restore protein homeostasis [44,45,46]. 

A fundamental problem, however, is that it is difficult to compare the various CAP studies with each other, as voltage, frequency, carrier gas (e.g., argon, helium), cell types, cell culture media, treatment type (direct vs. indirect), and experimental design often differ, which makes comparisons and interpretations difficult. 

### 3.2. Cell Signaling, Antioxidative Response, and Cellular Functions in CAP-Induced Oxidative Stress

As an important second messenger, Ca^2+^ regulates various cellular functions, including contraction, metabolism, gene expression, cell survival, and cell death [47]. In this context, it is crucial that oxidative stress be able to directly or indirectly affect calcium homeostasis. In our study, we demonstrated intracellular Ca^2+^ levels generally increased upon CAP-activated medium treatment (the increase was dependent on the CAP activation time and, thus, on the amount of CAP-induced ROS). The interaction between ROS and Ca^2+^ is known to be bidirectional, as increased Ca^2+^ concentrations activate ROS-generating enzymes and the formation of free radicals, which in turn can increase ROS production [48,49]. Furthermore, ROS can impair the activity of calcium channels, stimulate Ca^2+^ release from intracellular stores, and inhibit calcium pumps [50]. The calcium channel’s transient receptor potential (TRP), for example, which is activated via oxidative stress, causes an opening of the channels, leading to an influx of calcium ions into the cell [51,52,53,54]. The phospholipases, which hydrolyze the membrane phospholipids, are also activated and release the messenger substance inositol triphosphate-3, which stimulates the release of Ca^2+^ from intracellular stores, e.g., the endoplasmic reticulum [55,56]. 

In response to the CAP-induced oxidative stress, cells can activate signaling pathways that lead to the upregulation of various stress-associated proteins to maintain specific cell functions and, thus, cell survival [57,58]. In our study, 1-day treatment with a mildly CAP-activated medium (CAP-actM^1min^) increased antioxidant proteins in the adMSCs, such as carbonic anhydrase, PON2, SOD2, and thioredoxin-1. Treatment with a more strongly CAP-activated medium (i.e., CAP-actM^5min^), characterized by increased amounts of ROS, had the opposite effect: a reduction in stress-associated proteins. Since a number of the antioxidant stress-associated proteins investigated in this study, such as carbonic anhydrase, PON2, SOD2, and thioredoxin-1, are also involved in the regulation of Ca^2+^, it is debatable which occurs first, the calcium signaling and the subsequent increase in stress-associated proteins or vice versa. The relationship between this signaling is particularly complex, as calcium signaling is not solely responsible because even under the intensified CAP conditions, the Ca^2+^ level increases permanently, while the protein quantity of the stress-associated protein investigated subsequently decreases in adMSCs in this study.

The enzyme carbonic anhydrase, which was found to be affected by the CAP-activated medium in our study, is involved in the conversion of carbon dioxide and water to carbonic acid and, thus, in the pH regulation of the cell [59,60,61]. Hence, the increase in carbonic anhydrase protein observed after CAP treatment indicates a change in the pH of the cells that can affect the activity of calcium channels and might be responsible for the increased intracellular Ca^2+^ level. PON2 is also an enzyme that has antioxidant properties and is involved in protecting cells from oxidative stress [62]. A PON2 deficiency or an impairment of its function can increase oxidative stress, which can affect calcium signaling [63]. SOD2 and thioredoxin-1 also play a role in the protection against oxidative stress: the enzyme SOD2 converts superoxide radicals into H_2_O_2_ and oxygen, while the reduced thioredoxin-1 has an antioxidant effect in enzymatic form as oxidoreductase [57]. A SOD2 deficiency or an impairment of thioredoxin-1 can lead to an increase in ROS, which in turn can impair Ca^2+^ regulation [64,65]. Human keratinocytes also showed an activation of SOD2 after treatment with the CAP-activated medium [66]. Thus, oxidative stress affects the cellular redox environment in different signaling pathways via enzymes such as carbonic anhydrase, PON2, SOD2, and thioredoxin-1 and can, therefore, influence calcium regulation and highlight the complex interactions between these different aspects. We can summarize that 1-day treatment with a mildly CAP-activated medium increases the number of proteins and enzymes responsible for the defense against oxidative stress. However, treatment with a strongly CAP-activated medium decreases these proteins in adMSC cultures in our in vitro approach, indicating a depletion in the oxidative stress response.

Cell migration (including MSC migration) is a crucial process in tissue regeneration to position the cells at the relevant sites in the respective tissue [67,68]. In our in vitro approach, cells generally showed increased migration after treatment with a CAP-activated medium. This increased migratory activity is accompanied by a CAP-induced increase in intracellular Ca^2+^ levels. Studies have shown that increased intracellular Ca^2+^ levels can be associated with increased adMSC migration [69]. So far, however, there are only a few studies to date that have investigated the influence of CAP on cell migration, and these results cannot be described as conclusive overall: Direct treatment with CAP to murine fibroblast or a human glioblastoma cell line reduced the migration of the respective cells [70,71]. This reduced migration is in contrast to our results and the results of murine models, which showed a positive effect of CAP treatment on skin wound closure, which is inevitably associated with epithelial cell migration [72,73]. However, it must be emphasized that the different experimental models were run under various conditions (e.g., selected cell type/animal model, cell culture conditions/culture medium, voltage, frequency, carrier gas, distance of the plasma source to cells/tissues).

There are various effects on the motility of cells that increased intracellular Ca^2+^ levels can induce: for example, the activity of adhesion molecules such as cadherins and integrins, which are involved in the formation of cell–matrix and cell–cell junctions whose activity is controlled via intracellular Ca^2+^ [74,75]. Calcium-dependent proteases, e.g., from the calpain family, can influence the organization of the cytoskeleton. Calpains are activated via increased intracellular Ca^2+^ and cleave various substrates, including cytoskeletal proteins such as actin and myosin [76,77]. This process, in turn, controls the formation dynamics of protrusions, which are essential for cell migration [78]. Therefore, cell migration is mediated via changes in Ca^2+^ levels controlled by various proteins and signaling pathways and can vary depending on cell type and physiological state [79]. Thus, regulating calcium homeostasis is responsible for maintaining multiple cell functions. Consequently, the dysregulation of calcium homeostasis can lead to pathological conditions. Therefore, these complex results require further in-depth analyses of the effects of CAP on cellular signaling. 

In summary, the treatment of adMSC cultures with a CAP-activated medium did not cause cytotoxic effects when the treatment was short-term and non-repetitive. The known antioxidant capacity of MSCs [23,69] is probably responsible for the low intracellular ROS levels after treatment with a CAP-activated medium. In addition, short-term treatment with a CAP-activated medium resulted in increased migration and increased intracellular Ca^2+^ levels. Also, short-term treatment with a mildly activated CAP medium allowed adMSCs to increase various cell stress proteins, apparently adapting to the oxidative stress induced by CAP activation. Consequently, the effects of CAP-induced reactive species depend on the composition and concentration of reactive species, the cell culture conditions, and the selected cell type [80,81]. Therefore, the results of this in vitro study could emphasize certain aspects that are also relevant for the therapeutic application of CAP. For example, the frequency of treatment and precise control of plasma parameters should not be underestimated to support tissue regeneration through increased cell migration and avoid undesired damage to the treated tissue.

## 4. Materials and Methods

### 4.1. Cell Isolation and Cultivation

Human adMSCs were isolated from the liposuction tissue of healthy donors. For this purpose, the so-called stromal vascular fraction (SVF) was enzymatically isolated from the liposuction tissue according to the previously described protocol [82]. Afterward, the CD34-positive subpopulation was specifically separated from the SVF via a magnetic bead-based protocol [83] and cryopreserved in passage 2. Cell thawing, passaging, and seeding for further experiments were carried out as previously described [84]. All experiments were carried out in passage 4. Cells were seeded at a density of 20,000 cells/cm^2^ in 4- to 96-well plates (Greiner Bio-one, Frickenhausen, Germany) unless otherwise stated. The absence of mycoplasma contamination was confirmed via the microscopic analysis of the absence of cytoplasmic DNA (staining with diamidino-2-phenylindole/DAPI, Sigma Aldrich, Saint Louis, MO, USA).

### 4.2. Plasma Source

The medical device kINPen^®^ Med (neoplas tools GmbH, Greifswald, Germany) was used to generate the cold atmospheric plasma, which has been characterized concerning physical and biological parameters following DIN Spec91315 [27]. The design and function were previously described in detail [85]. The device generated a radio frequency signal of approximately 1 MHz and a voltage amplitude of 2–3 kV. The argon flow rate was set to 5 standard liters per minute, with the plasma being generated at the tip of the pin electrode and spreading into the ambient air outside the capillary. The media were treated directly at a distance of approximately 12 mm from the capillary outlet to the surface. Therefore, the argon plasma had direct and consistent contact with the cell culture medium.

### 4.3. Generation of CAP-Activated Cell Culture Medium

To activate the cell culture medium with CAP, 5 mL of medium were exposed to argon plasma in a 6-well plate for 0 min (Ctrl.), 1 min (CAP-actM^1min^), 3 min (CAP-actM^3min^), and 5 min (CAP-actM^5min^), which we refer to as CAP-activated medium in the following section. The control was only exposed to argon gas without igniting the plasma. Two different cell culture medium compositions were used for the experiments to provide different oxidative conditions: the standard Dulbecco’s Modified Eagle Medium (DMEM: PAN Biotech, Aidenbach, Germany), which contains 110 mg/L of the ROS scavenger sodium pyruvate (referred to as DMEM^pyr+^) and a sodium pyruvate-free DMEM (DMEM^w/o pyr^; Gibco by Life Technologies, Paisley, UK). In principle, all experiments were conducted in a medium supplemented with 10% fetal bovine serum (PAN Biotech, Aidenbach, Germany), 1% penicillin/streptomycin, and 0.4% GlutaMax™ (Gibco, by Life Technologies, Darmstadt, Germany).

Immediately after plasma activation, the CAP-activated medium was added directly to the pre-cultured adMSCs once for cultivation until day 1, twice for cultivation until day 3, or thrice for cultivation until day 7. Treatment with CAP-activated media took place with media changes every 2 days.

### 4.4. Quantification of Extracellular H_2_O_2_ and Intracellular ROS Level

The concentration of H_2_O_2_ produced via the CAP activation of the medium was measured using the Amplex^®^ UltraRed assay (Thermo Fischer Scientific, Waltham, MA, USA). The assay was performed according to the manufacturer’s instructions. The CAP-activated media were diluted with phosphate-buffered saline (PBS, Ca^2+^- and Mg^2+^-free, PAN, Aidenbach, Germany) to achieve the linear detection range of the assays. The measurement was performed 15 min to 24 h after the addition of Amplex^®^ UltraRed. The relative fluorescence intensity was measured using a microplate reader (TECAN, Männedorf, Switzerland) at an excitation/emission wavelength of 530/590 nm. The H_2_O_2_ concentration was calculated using a standard curve.

Intracellular ROS were detected using the ROS indicator 2’-7’-dichlorofluorescein diacetate (CM-H2DCFDA, Thermo Fischer Scientific) according to the manufacturer’s instructions. For this purpose, CM-H2DCFDA was added to the adMSCs 72 h after seeding with a final concentration of 10 µM. A cell culture incubated with 100 µM of H_2_O_2_ served as a positive control. The experiment was performed according to an established protocol [69]. Cell cultures were treated with CAP-activated medium (DMEM^pyr+^ or DMEM^w/o pyr^) and control (non-CAP-activated) medium for up to 60 min. Fluorescence was quantified with a fluorescence microplate reader (TECAN, Männedorf, Switzerland) every 10 min at 492 nm excitation and 527 nm emission until saturation was reached. The results were normalized to the cell number to estimate the amount of accumulated ROS per cell.

### 4.5. Determination of Cell Number

The adMSC cell number was quantified with the nuclear dye Hoechst 33342 (Applichem, Darmstadt, Germany) after 1, 3, and 7 days. The quantification of cells was performed at the level of cell nucleus count according to an established protocol [69]. The resulting cell number was normalized according to the untreated control cultures.

### 4.6. Analysis of Metabolic Activity

The metabolic activity of adMSCs was determined with the CellTiter 96 Aqueous One Solution Cell Proliferation Assay (MTS, Promega, Madison, WI, USA) after 1, 3, and 7 days of treatment with CAP-activated medium, according to the manufacturer’s instructions. The incubation and measurement were carried out based on the previously described protocol [83]. The viability was calculated per cell number and normalized to untreated control cultures. 

### 4.7. Quantification of Mitochondrial Respiration and Extracellular Acidification

The Agilent Seahorse XFp Cell Mito Stress Test (Agilent Technologies, Santa Clara, USA) was used to analyze mitochondrial respiration and glycolysis. An assessment of mitochondrial and glycolytic activity is possible by directly measuring the oxygen consumption rate and the extracellular acidification rate. The respiratory chain modulators are oligomycin, which inhibits the complex V (ATP synthase) of the respiratory chain; carbonyl-cyanide-4-(trifluoromethoxy)-phenylhydrazone (FCCP), an uncoupling agent that disrupts the proton gradient and mitochondrial membrane potential; and a complex of rotenone and antimycin A, inhibitors of complex I and complex III. For the analyses, adMSCs were seeded in an 8-well ‘miniplate’ at a density of 15,000 cells/well according to the manufacturer’s instructions. After 3 days of pre-cultivation, adMSCs were incubated with CAP-activated medium for 1 and 3 days. For the measurement itself, the CAP-activated medium was replaced with DMEM containing 1 mM of pyruvate, 2 mM of glutamine, and 10 mM of glucose at a pH of 7.4. Unless otherwise stated, all reagents and plastic devices were from Agilent Technologies (Agilent Technologies, Santa Clara, CA, USA).

### 4.8. Measurement of Intracellular Basal Ca^2+^ Level

The intracellular basal Ca^2+^ levels were determined via flow cytometry after specific labeling with the calcium indicator Fluo-3-acetoxymethyl ester AM. For this purpose, adMSCs were first seeded in a 24-well plate, pre-cultivated for 3 days, and treated for 1 day with a single or for 3 days with a double application of CAP-activated media. The experiment was performed according to an established protocol [86]. Intracellular basal Ca^2+^ levels were analyzed using the FACSCalibur (Becton Dickinson, BD Biosciences, Franklin Lakes, NJ, USA) flow cytometer equipped with an argon laser (λ 488 nm) and CellQuest Pro 4.0.1 software (BD Biosciences). For the data acquisition, FL1-H was determined using FlowJo_V.10.1r1 (FlowJo, LLC; BD Becton Dickinson and Company, Franklin Lakes, NJ, USA). 

### 4.9. Quantification of Cell Stress-Associated Proteins

Cell stress-related proteins were detected using the human cell stress array kit (R & D Systems, Minneapolis, MN, USA). For the cell lysate isolation required for this, adMSCs were treated with CAP-activated medium (DMEM^W/O **pyr**^). After one day of treatment with a CAP-activated medium, the cells were prepared as described in a previously established protocol [69]. After that, cell stress-related proteins (Figure A2) were detected using chemiluminescence blots, which were recorded with ChemiDoc XRS (Bio-Rad Laboratories GmbH, Munich, Germany), and protein intensity was quantified via a densitometry analysis of the blot using the Image Lab 3.0.1 software (Bio-Rad, Laboratories GmbH, Munich, Germany).

### 4.10. Determination of Migration Activity

To evaluate the effect of CAP-activated media on the migratory activity of adMSCs, we performed the so-called wound closure assay (also referred to as scratch assay). For this purpose, cells were seeded at 29,000 cells/cm^2^ in a 96-well plate to achieve an almost 100% confluent monolayer. One day after seeding, cells were stained with a red DMSO-free cell-tracking dye (Abcam, Cambridge, UK) according to the manufacturer’s instructions. When 3 days after seeding was reached, the medium was replaced with CAP-activated medium (DMEM^+pyr^ or DMEM^w/o pyr^) at the indicated time points either once for 1 day or twice for 3 days. Subsequently, the cells were scratched from the confluent monolayer. The cells were washed 3 times with PBS and treated with the different CAP-activated media. Live-cell imaging was observed immediately after scratching via the Hermes WiScan system (IDEA Bio-Medical, Rehovot, Israel) at 10-fold magnification in a humidified atmosphere with 37 °C and 5% CO_2_ for 15 h. The quantification of the wound closure and, thus, the migratory activity of the MSCs was performed using Fiji ImageJ software version 1.8.0_72 [87], and the percentage of wound closure was calculated by the following formula:(initial wound area − final wound area)/initial wound area × 100%

### 4.11. Statistical Analysis

All experiments were performed independently on adMSCs from four donors, each in triplicate, and the mean of each triplicate was used for one individual. The data were visualized and statistically analyzed with Microsoft Excel 2010 and GraphPad Prism, Version 7 (GraphPad Software Inc, San Diego, CA, USA). The data were presented as bars with standard deviations and boxplots. The horizontal line within the box plot indicates the median and mean (+), with whiskers showing the minimum and maximum data points. The Shapiro–Wilk test indicated Gaussian distribution; therefore, the statistical significance of the dataset was calculated using ordinary one-way or two-way analysis of variance, followed by Dunnett’s multiple comparison tests, as appropriate to the specific issue, with the significance level set at a *p* value of 0.05.

## 5. Conclusions

The interactions between oxidative stress, the activation of stress-related proteins, and increased cell migration are complex and can vary in different physiological and pathological situations. Activating stress-related proteins via oxidative stress may be an adaptive response that induces MSCs to adapt to their environment. In contrast, excessive stress can lead to severe cell damage. Therefore, the results of this in vitro study could highlight certain aspects that are also relevant for therapeutic CAP application. For example, the treatment frequency and the precise control of plasma parameters should be considered in order to provide regenerative support and avoid undesirable side effects. Thus, these results contribute to the knowledge of the interface between CAP treatments and wound healing by providing a deeper insight into the physiological responses of MSCs during indirect CAP application.

## Figures and Tables

**Figure 1 ijms-25-04944-f001:**
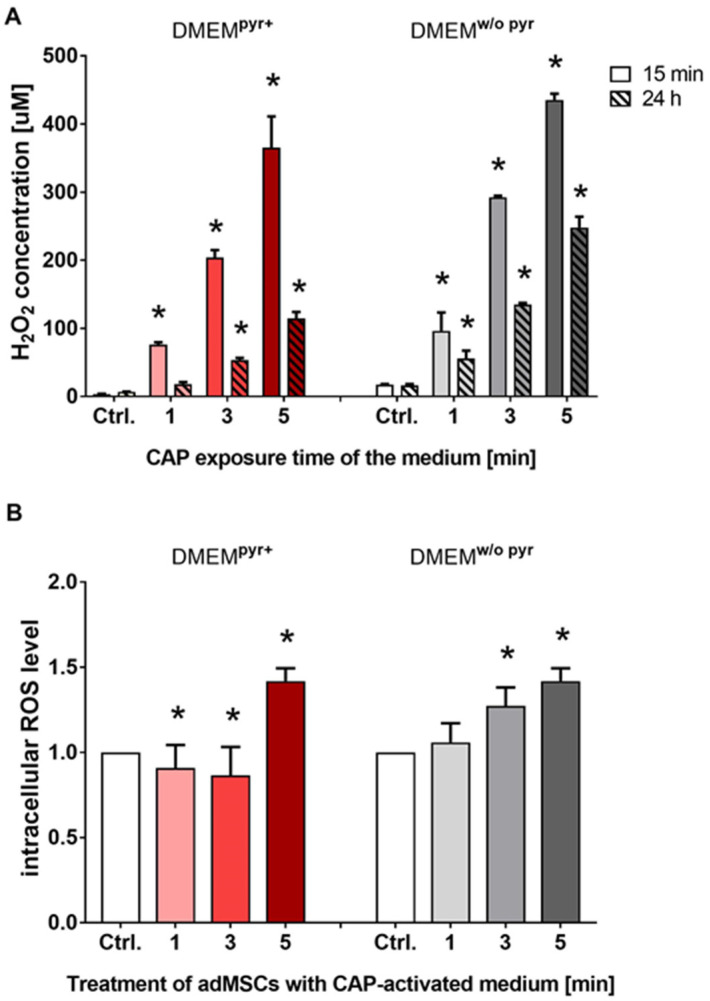
Quantification of ROS concentration in cell culture medium after CAP treatment (**A**) and after treatment of cells with CAP-activated medium (**B**). (**A**) Quantification of H_2_O_2_ in CAP-activated medium (i.e., untreated control medium/Crtl., CAP-actM^1min^, CAP-actM^3min^, and CAP-actM^5min^ in the presence of the ROS scavenger pyruvate (DMEM^pyr+^, **left**) and the absence of pyruvate (DMEM^w/o pyr^, **right**) after 15 min and 24 h from the start of the quantification (measured via Amplex UltraRed assay). (**B**) Determination of intracellular ROS level in adMSCs after 30 min of treatment with CAP-activated medium using a fluorescence microplate reader (relative intracellular ROS level measured using CM-H2DCFDA); (means with standard deviations, normal distribution was proven using the Shapiro–Wilk test, ordinary one-way ANOVA with Dunnett’s multiple comparison post hoc test; * *p* < 0.05 significant compared to the control, n = 4).

**Figure 2 ijms-25-04944-f002:**
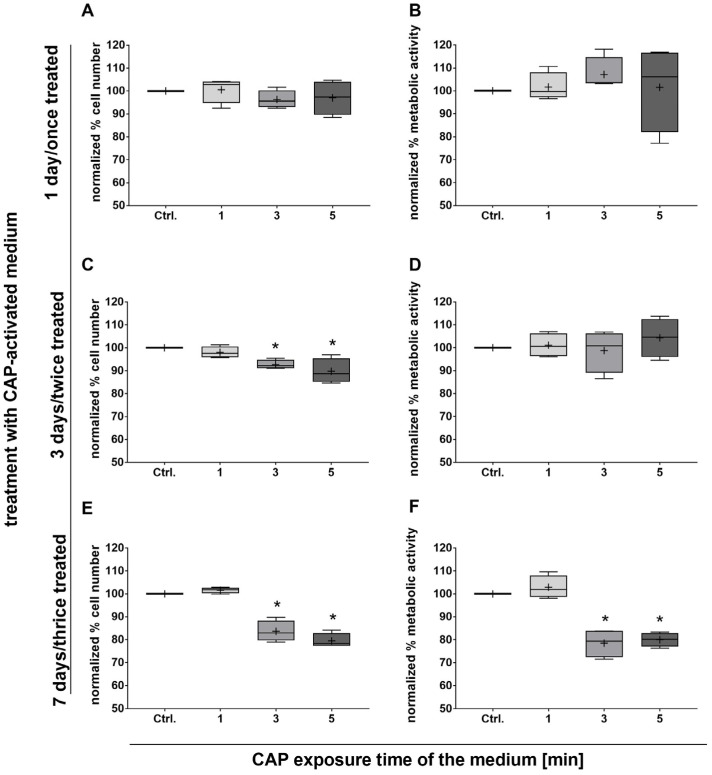
The effect of CAP-activated pyruvate-free medium on adMSC number and metabolic activity. adMSC numbers (**A**,**C**,**E**) and metabolic activity (**B**,**D**,**F**) were quantified after 1 day of single treatment (**A**,**B**), after 3 days with double treatment (**C**,**D**), and after 7 days with triple treatment with CAP-activated media (**E**,**F**). Cell numbers were analyzed through the quantification of cell nuclei (Hoechst 33342), and metabolic activity was quantified through a tetrazolium salt assay (dataset was normalized to the control; the Shapiro–Wilk test indicated normal data distribution; statistical significance was calculated using an ordinary one-way ANOVA with Dunnett’s multiple comparison post hoc test; * *p* < 0.05 significant compared to the control, n = 4).

**Figure 3 ijms-25-04944-f003:**
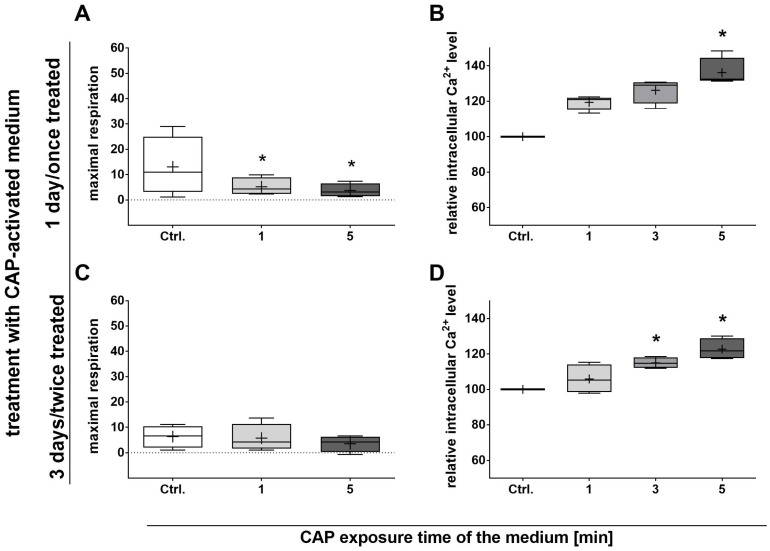
The effect of CAP-activated pyruvate-free medium on adMSC mitochondrial respiration and intracellular Ca^2+^ level. Maximal mitochondrial respiration (**A**,**C**) and basal intracellular Ca^2+^ level (**B**,**D**) were quantified after 1 day with a single and 3 days with a double treatment of adMSCs with CAP-activated medium. Mitochondrial respiration was measured using the relative oxygen consumption rate ((pmol/min)/5 × 10^6^ cells/mL), and detection of basal intracellular Ca^2+^ level was carried out with a calcium indicator with flow cytometry (fluorescence intensity at 488 nm; dataset was normalized to the control; Shapiro–Wilk test indicated normal data distribution; statistical significance was calculated using an ordinary one-way ANOVA with Dunnett’s multiple comparison post hoc test; * *p* < 0.05 significant compared to the control, n = 4).

**Figure 4 ijms-25-04944-f004:**
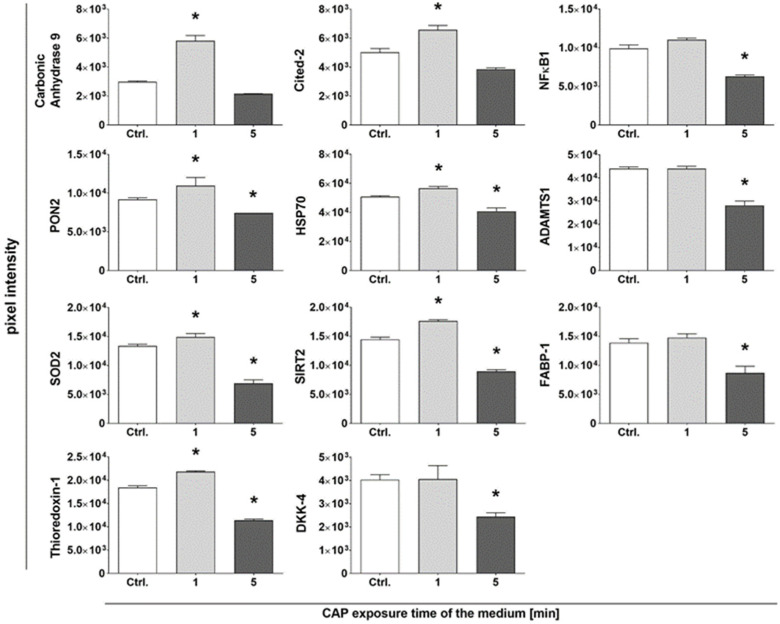
Analysis of human cell stress proteins in adMSCs after 1 day, i.e., after a single treatment with CAP-activated medium. Proteins associated with oxidative stress or cytoprotection were detected using the ‘Proteome Profiler Human Cell Stress Array Kit’ and quantified with densitometry analysis (pixel intensity of the protein spots) using the Image Lab 3.0.1 software (Shapiro–Wilk test indicated Gaussian distribution; statistical significance was calculated using an ordinary one-way ANOVA with Dunnett’s multiple comparison post hoc test; * *p* < 0.05 significant compared to the control, n = 4).

**Figure 5 ijms-25-04944-f005:**
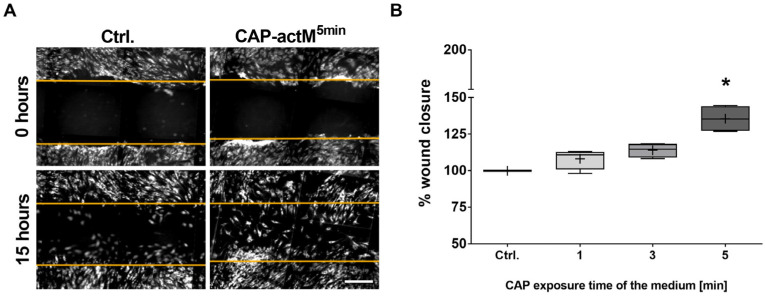
Analysis of the migratory activity of adMSCs after 1 day with a single treatment with CAP-activated medium. (**A**) Representative microscopic images of untreated adMSCs and adMSCs treated with CAP-activated medium immediately after scratching (i.e., at 0 h) and 15 h thereafter (the orange lines mark the respective boundaries of the original scratch area). (**B**) Quantification of wound closure via image analyses (quantification of cell overgrowth was 15 h after scratching, scale bar: 100 µM; Shapiro–Wilk test indicated Gaussian distribution; statistical significance was calculated using a two-way ANOVA with Dunnett’s multiple comparison post hoc test; * *p* < 0.05 significant compared to the control, n = 4).

## Data Availability

The data presented in this study are available in the article, and from the corresponding author.

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
