# Peer review of "Cold Atmospheric Pressure Plasma-Activated Medium Modulates Cellular Functions of Human Mesenchymal Stem/Stromal Cells In Vitro"

_ijms, 2024, doi:10.3390/ijms25094944_

Round 1

Reviewer 1 Report

Comments and Suggestions for Authors

This study examines the impact of cold atmospheric pressure plasma (CAP) on mesenchymal stem/stromal cells (MSC). CAP activates oxidative stress-related reactive species in the culture medium. Single short-term exposure increases intracellular calcium ion levels and migration rates of MSC, alongside elevated stress proteins, suggesting adaptation to oxidative stress. However, repeated exposure proves cytotoxic. The authors also conclude that the CAP-induced reactive species exhibit both beneficial and detrimental effects on MSC. The manuscript required major modifications before publication.

1. What is the novelty of this work? It should be highlighted in the abstract section.

2. As the field of plasma is broad, the background information provided by the authors may be insufficient. The following recent article could be a useful resource to incorporate into the background information. [Review on the Biomedical and Environmental Applications of Nonthermal Plasma. Catalysts 2023, 13.]

3. Why do authors prefer to use argon gas by knowing the fact the ROS concentration will be high which might be harmful for normal cells at 5mint treatment? Why not RNS?

4. In line 93, “52–57% of this initial value” what does it mean initial concentration? It should be reduced to the concentration that is observed after 1 minute of treatment.

5. In the Figure 1 caption, replace ‘characterization’ with ‘concentration’ and CAP activation with CAP treatment.

6. H2O2 is long-lived species. Why did H2O2 concentration dramatically decrease after 24h incubation? Explain the reason in detail.

7. How is the concentration of ROS measured in Figure 1B? Are these intracellular ROS levels or inside media?

8. The H2O2 concentration reached approximately 500 μM after 5 minutes of treatment, posing a potential stress threat to cells. Surprisingly, there was no discernible alteration in cellular metabolic activity following a single treatment. Explain the reason.

Also, Considering the subsequent decrease in H2O2 levels after 24 hours, what is the time gap between 2nd and 3rd treatments?

9. In Figure 5A, draw the reference line. It is hardly visible without a line.

10. In Figure 5A, are the presented results of the migration assay after a single treatment, the second treatment, or the third treatment?

11. The authors should enhance the quality of their discussions by providing more detailed descriptions of the key findings and elucidating the underlying mechanisms or reasoning behind the observed effects. In the present form, I am unable to find these details.

12. The author concludes the study poorly, most of the information is known and common sense. I recommend rewriting the conclusion, particularly describing the key findings of the presented study.

13. The authors should add analysis beyond ROS and include measurements of pH, ORP (oxidation-reduction potential), EC (electrical conductivity), and other species such as reactive nitrogen species (RNS) in the media post-CAP treatment. This comprehensive approach will provide a more complete understanding of the effects induced by CAP treatment. Is there any specific reason to focus on ROS only?

14. I recommend thoroughly reviewing the manuscript to identify and rectify any typos and grammatical errors.

Comments on the Quality of English Language

I recommend thoroughly reviewing the manuscript to identify and rectify any typos and grammatical errors.

Reviewer 2 Report

Comments and Suggestions for Authors

The manuscript is based on the effect of two cold-activated pressure media on human mesenchymal cell peroxide formation, migration, and calcium metabolism. The article has a clear rationale, but the results need to be expressed better to be easier for the reader to interpret. Figure 1 is confusing; it would be better to stick with the plain white stripes for the two conditions in part A of the figure or change the bars to lines. Please make the * for significant difference clear in the figures. The difference is evident after 24 hr incubation, but the range of colours confuses the reader. Figure 1 B should be discussed further since there seems to be a lag period of three minutes between the two conditions, which may affect the interpretation of Figures 3C and D. I am confused with Figures 2 A and C as to why the normalised cell number differs in 5 min. Then, the figure should be changed to ratios. Figure 4 is also confusing; the increase and decrease of carbonic anhydrase is very high for a short period. Also, SOD2 at 5 min.  It is very difficult to conceive a small SD in pixel analysis.  Figure 5  should be modified. Part A does not show a significant migration after 15 hrs. In part B, significant differences, at 3 and 5 min, do not exist for an n=4. The discussion is based on assumptions which are not as described.

Comments on the Quality of English Language

Several minor grammatical mistakes were encountered

Round 2

Reviewer 1 Report

Comments and Suggestions for Authors

I appreciate that the authors have made significant and suitable modifications in the revised version. But there are still two matters that need to be addressed carefully before publication.

1. If the migration assay, conducted on day 3 after the double treatment, shows a slight increase in cell migration, it is important to note that previous results indicated the double treatment have harmful effects on cells. Therefore, comparing these results with a control group (not treated with anything) is not the appropriate experiment. Instead, the author should compare the outcomes of the double treatment with those of the single treatment, which is the primary focus of this research.

2. In Figure 1, the author mentioned that these are intracellular ROS levels measured by CM-H2DCFDA. It is essential to include fluorescent images depicting the ROS levels alongside quantification of the ROS intensity. This is necessary for proper presentation and clarity; quantitative results alone without accompanying images are not sufficient or acceptable.

Reviewer 2 Report

Comments and Suggestions for Authors

The manuscript was corrected accordingly, as a pilot study it can be published.

Comments on the Quality of English Language

Minor grammatical mistakes were encountered in the new text

Round 3

Reviewer 1 Report

Comments and Suggestions for Authors

I recommend accepting the paper in its present form.